# Aquatic Biota Is Not Exempt from Coronavirus Infections: An Overview

Gabriel Núñez-Nogueira [1,*], Jesús Alberto Valentino-Álvarez [1], Andrés Arturo Granados-Berber [1], Eduardo Ramírez-Ayala [2,3], Francisco Alberto Zepeda-González [2] and Adrián Tintos-Gómez [2,3,*]

1  Hydrobiology and Pollution Laboratory, DACBiol Juarez Autonomous University of Tabasco, Carretera Villahermosa-Cárdenas Km. 0.5, Villahermosa 86039, Tabasco, Mexico; jesusalberto.valentinoalvarez@gmail.com (J.A.V.-Á.); andres.granados@ujat.mx (A.A.G.-B.)
2  Department of Studies for the Development of the Coastal Zone, University of Guadalajara, Guadalajara 48900, Jalisco, Mexico; eduardo-ramirez@utem.edu.mx (E.R.-A.); francisco-zepeda@utem.edu.mx (F.A.Z.-G.)
3  Renewable Energy Research Centre, Technical Secretariat of the Academic Area, Manzanillo University of Technology, Manzanillo 28869, Colima, Mexico
*  Correspondence: gabriel.nunez@ujat.mx (G.N.-N.); adrian-tintos@utem.edu.mx (A.T.-G.)

**Abstract:** Coronaviruses are pathogens recognized for having an animal origin, commonly associated with terrestrial environments. However, in a few cases, there are reports of their presence in aquatic organisms like fish, frogs, waterfowl, and marine mammals. None of these cases has led to human health effects when contact with these infected organisms has taken place, whether they were alive or dead. Aquatic birds seem to be the main group carrying and circulating these types of viruses among healthy bird populations. Although the route of infection for COVID-19 by water or aquatic organisms has not yet been observed in the wild, the relevance of its study is highlighted because there are cases of other viral infections known to have been transferred to humans by aquatic biota. It is encouraging to know that aquatic species, such as fish, marine mammals, and amphibians, show very few coronavirus cases. Some other aquatic animals may also be a possible source of cure or treatment against, as some evidence with algae and aquatic invertebrates suggest.

**Keywords:** coronavirus; aquatic organisms; fish; marine mammals; frogs; birds

## 1. Introduction

The current Severe Acute Respiratory Syndrome coronavirus 2 (SARS-CoV-2) or COVID-19 pandemic, as it is commonly known, brought society's interest in one of the families of high-risk pathological-infectious viruses, known as Coronavirus (CoVs) [1]. The impact that COVID-19 has had on human dynamics is undoubtedly enormous. The mortality and public health impacts caused by COVID-19 caught the attention of scientists to try to slow down its effects and look for a vaccine. This virus is already present in every continent, and as with previous events with other viruses such as SARS or HIV (Human Immunodeficiency Virus), humans will have to learn how to live with it. However, this situation also makes us wonder about what other organisms may be subject to coronavirus infection. Which organisms can be vectors or reservoirs? They may have the virus in their body, transport it, and spread it in other areas or to other organisms without suffering the symptoms of the infection [2].

Moreover, can the coronavirus infect and affect aquatic organisms? Indeed, these questions in the scientific and non-scientific communities may eventually be answered in a particular way over time for SARS-CoV-2. However, at the moment, scientific efforts are focused on the public health aspects at the global level [3,4]. Thanks to previous studies on the subject, we can access information to understand more about the possible scenarios associated with these questions. In addition, it allows us to be able to make more specific approaches to the impact of SARS-CoV-2 on aquatic organisms, based on

the general knowledge that is available on coronaviruses. Due to the above, this review aims to synthesize the information available in the scientific literature on the detection and presence of coronavirus in organisms and promote its study, not only for the ecological and public health impacts, but also its potential as treatment sources. This work has focused on the coronavirus' basic features, presence in aquatic environments, detection in aquatic organisms (fish, marine mammals, waterfowls, amphibians, crustaceans, and mollusks), some viral infections on humans from aquatic organisms, and further considerations involving biological aspects or potential use of biomolecules produced by aquatic biota, in search of a treatment or control against coronavirus exposures. This review highlights the proven CoVs cases, discussing if their features could indicate future CoVs infections, taking into account previous infections from similar viruses, as well as the possible implications in its dispersion. It also mentions the potential that aquatic organisms can represent in searching for control or treatment against these highly pathogenic viruses. The importance of carrying out broad and specific studies on coronavirus presence, abundance, pathologies, dispersion, and affectations in aquatic biota is discussed.

This review was achieved by mainly identifying formal research articles published and available in different scientific databases such as Web of Science, Scopus, PubMed, Google Academic, ResearchGate, and ScienceDirect. The studies included in this review were based on the following selection criteria: studies reporting the presence of coronavirus in aquatic organisms, including amphibians and birds. Those published in peer-reviewed journals were considered preferably due to the limitation of works. Search-engines included "coronavirus", "SARS", "CoV", "fish", "crustacea", "marine mammals", "birds", "waterfowl", "amphibian", "treatment", "source", "substance", "infection", "disease ", "host", "health", "mollusk "," invertebrates "," marine", "freshwater ", "aquatic "," biota "," organism ", and their combinations. General information was collected for every eligible study, including author(s), year published, coronavirus type, order, family, gender, species, and host. The adverse or health effect was also recorded when possible, specifically whether this concerned coronavirus or a related virus.

*Coronavirus Features*

CoVs are pathogens associated with epithelial cell infections such as gastrointestinal (gastroenteritis) and respiratory (respiratory infections) [5–7]. Its structure consists of three components: (1) genetic material with which it replicates or reproduces within infected cells, known as RNA (ribonucleic acid of a chain between 26 and 32 kb in length), (2) an external protein that surrounds it known as caps (viral wrap), and (3) a membrane that surrounds and envelops the protein cover, which is covered, in turn, with spicules that give the shape of a "crown", from which they are called coronavirus, and that allows them to recognize and come into contact with the membrane of the cell that will be infected [8,9]. These spicules are called the "S" protein [8].

It is recognized that CoVs have their origin in bats, with several varieties or viral species depending on poly-protein or full genome analyses [9,10], which include some of the most toxic and lethal strains of recent decades. CoVs mainly affect terrestrial organisms, such as humans, bats, felines, camels, and birds [9,11]; however, their potential to infect aquatic life has been demonstrated (Figure 1).

Coronaviruses are part of the Family Coronaviridae of the Order Nidovirals, which in addition to infecting terrestrial mammals and birds, it has also been found in frogs, fish, and marine mammals [7,9,12]. Four genera are recognized based on their phylogeny and genomic structure, from the subfamily Orthocoronavirinae, of which *Alphacoronavirus*, *Betacoronavirus*, *Gammacoronavirus*, and *Deltacoronavirus* stand out, for their ability to infect humans and non-human respiratory tracts, and other organisms at the digestive level (enteritis) [9]. Within this group of Nidovirals, we find viruses such as the widely known SARS-CoV and MERS-CoV (Middle East Respiratory Syndrome), both belonging to the genus *Betacoronavirus* and subgenus *Sarbecovirus* and *Merbecovirus*, respectively [7,9]. These two types of viruses are recognized as infectious of zoonotic origin, implying a transmission

from an animal due to direct interactions with an animal carrying the infection. The other subfamilies within the Coronaviridae are the Torovirinae, formed in turn by two genera, *Torovirus* and *Bafinivirus*, and Letovirinae with *Alphaletovirus* (LeV), respectively. *Bafinivirus* has been identified in a teleost fish [13,14] and LeV in a frog [12], a clear example that it is possible to find coronaviruses in aquatic environments.

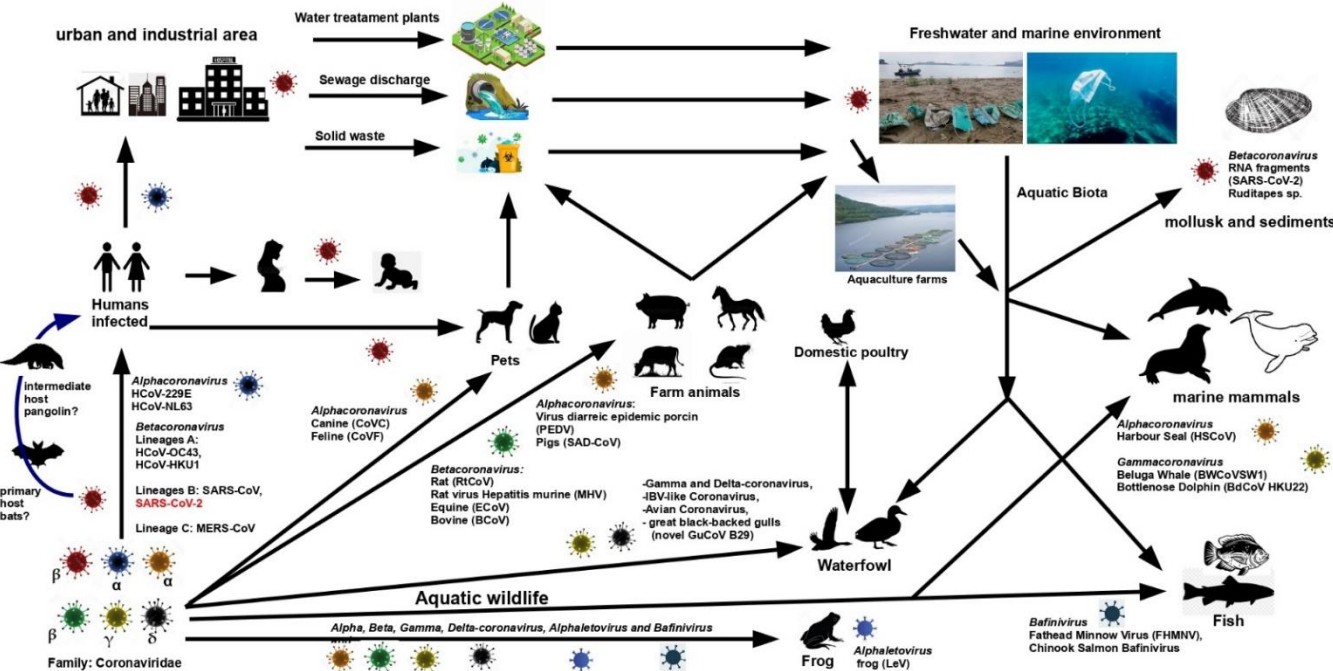

**Figure 1.** Main coronavirus groups and routes of direct or indirect transfer towards aquatic environments and aquatic biota.

## 2. Coronavirus and Aquatic Environments

CoVs in natural waters have been determined with low percentages compared to other types of viruses; however, they have been detected recently, as are the cases in America, Asia, Australia, and Europe [15,16].

Interest in virus transmissions in aquatic media focuses on those with public health relevance, often entering natural water from treated and untreated wastewater discharges [5,17]. In the 1980s, it was recognized that as far as pathogenic viruses are concerned, more than 100 different types are excreted by man or by animals through their feces [18]. The stool is one of the main materials for virus transfers to water bodies. The transfer to the mouth as a result of poor hygiene, or the intake of contaminated water, allows viruses to enter the digestive system, infecting and replicating itself in the gastrointestinal tract, and then being expelled in large numbers again in the feces produced by infected people or animals [19]. It is estimated that about 10 billion viral particles are present for every gram of excrement [20]. Sewage, especially in countries with limited capacities for treatment and adequate sanitization, poses a risk of contamination when discharged or overflow occurs into bodies of natural water [20]. Unfortunately, this scenario becomes more complicated considering that hospital waste and biological-infectious and sanitary waste (intimate towels, mouth covers, gauze, body protectors, among others) can eventually reach lakes, lagoons, rivers, and seas, due to their mishandling as solid waste. They are becoming vectors of viruses in the aquatic environment. Recently, some studies have shown the presence of SARS-CoV-2 from the waste discharge activities of infected communities [16,21]. The potential for increased CoV survival in wastewater coincides with a period of increasing coronavirus infections in the human population [22]. Wastewater's physicochemical and biological parameters might influence the CoV survival capacity (e.g., temperature, pH, organic matter composition and content). An increase in temperature might be a crucial factor due to the sensitivity of

coronaviruses, considering that the wastewater temperature ranges from 10 °C in winter to more than 20 °C in the summer in some regions.

It should be noted that untreated wastewater is a source of microorganisms (e.g., bacteria), which can, at least to some extent, decrease the presence of viable viruses [23]. Detection of viral RNA does not equate to infectivity, as clearly shown in a study in which SARS-CoV genomic material was detected in untreated hospital wastewater. However, it was shown not to be viable in an in vitro cell line model [24].

On the other hand, for Casanova et al. [25], wastewater, including domestic and hospital wastewater, cannot be expected to represent a viable viral material transport route to an aquatic environment. However, observations of selected coronaviruses should be used with caution to predict the survival of untested species in this regard. As demonstrated, for TGEV (transmissible gastroenteritis virus) and MHV (mouse hepatitis virus), viruses required 9 and 7 days, respectively, until their infectivity in pasteurized sedimented wastewater was reduced down to 99% [25].

### 2.1. Coronavirus in Natural Waters

In general, the survival of coronaviruses in natural waters is likely to depend on four conditions: (i) water temperature; (ii) availability of light; (iii) level of organic matter; and (iv) predation [26]. Higher temperature reduces the survivability of enveloped RNA viruses [27,28]. Therefore, the extracellular survival of coronavirus in lakes and rivers will differ depending on geographic location, with potentially greater persistence in temperate areas compared to subtropical and tropical areas [29]. Depth should also impact survival, as shallow aquatic ecosystems tend to have higher mean water temperatures and little or no stratification [30].

Coronavirus' undetermined particles might undergo considerable breakdown and lose infectivity after reaching aquatic habitats. These viral-type particles include both eukaryotic and phage viruses, which undergo decay of particles in seawater at rates of 2–4% $h^{-1}$ [31], which is generally considered as a result of sunlight (UV-C radiation) [32,33]. Particular viral species may vary in susceptibility to ultraviolet light (UV type A, B, and C radiation). For example, UV-C light has been shown to cause a rapid and significant decrease in infectious SAR-CoV while it has not had the same effect on the canine coronavirus, the other representative of β-CoV, despite exposure of up to 3 days [34]. Thus, exposure to ultraviolet light can also decrease viral activity [35], so different effects of UV light on coronaviruses can be expected.

Viral particle adsorption on the suspended organic matter can, in turn, protect from light and affect settling behavior. It can also influence the viral diffusion coefficient [36] and potentially lead to virus clusters, particularly in waters with high suspended solids. The presence of antagonist microorganisms can also modify the survival of the viruses by inactivating CoVs in water. Some protozoa are known to feed on viruses, and some antiviral factors can be released by algae and actinomycetes, while extracellular bacterial enzymes can effectively inactivate selected viruses [23]. Also, the interaction with thermolabile organic matter can include nucleases and proteases present in marine microorganisms [37].

Wartecki and Rzymski [38] have reviewed the effect of temperature on the survival of some CoVs, where natural environmental conditions were partially imitated, finding variability in the infective ability of their hosts for different viruses and under different temperature conditions, as previously mentioned for wastewaters. For example, SARS-CoV only persisted for two days in tap water without chlorine at 20 °C but persisted for at least 14 days (the endpoint of the study) at 4 °C [26]. It should be noted that the only experimental study so far on the survival of coronavirus in lake water used samples taken from a reservoir that serves as a source of drinking water [23]. Therefore, it may not fully reflect survival in fresh surface waters that show significant differences in chemical and biological characteristics. More studies on survival under realistic conditions are required to understand the survival dynamics of coronaviruses in aquatic ecosystems, characterized by differences in temperature, organic matter availability, pH, and trophic status [24].

The marine environment can be contaminated by the flow of treated or untreated sewage or by bathers providing their body fluids [39,40]. In several studies, the SARS-CoV-2 genome has been detected in the marine environment or wastewater, but this does not mean that the detected virus is infectious [16,40,41]. Therefore, when alternative transmission routes are mentioned, factors that affect virus infectivity in the marine environment should be examined [39].

Salinity and pH are important factors affecting the survival and infectivity of viruses in marine conditions [40]. High concentrations of salt (especially NaCl) in water can significantly inactivate many viruses; thus, removing the salts from the solutions increases the survival time of the viruses [42].

The pH of the marine environment can affect the viral survival rate directly or indirectly. The direct effect of pH affects the conformation of the viral capsid protein and can affect viral survival. Studies carried out on HCoV 229E, MHV (murine hepatitis virus), and TGEV (transmissible gastroenteritis virus) showed a significantly faster inactivation in alkaline conditions (pH > 8) and longer viral persistence in slightly acidic conditions (pH = 6.0–6.5) [39,43].

The mean level of ocean salinity is around 35 parts per thousand (ppt or g $L^{-1}$), and the average pH is 8.1. The salinity-pH relationship shows that the pH of seawater increases with an increase in salinity; for example, in the Mediterranean Sea, salinity increases above 38 ppt, while pH increases to 8.5 [39,44–46]. According to Seyer and Şanlıdağ [38], due to the great dilution capacity of the ocean and the characteristics of seawater such as high salinity and pH, viral inactivation can occur considering an infectious dose of SARS-CoV-2 (~300). Thus, in marine environments, they are not likely to be a risk factor for the transmission of SARS-CoV-2 [38]. However, it is necessary to continue evaluating how the physicochemical factors, like the acidity of the seas, salinity, and osmotic pressure, might affect the viruses' interactions with populations of marine-coastal species and humans in the immediate future. These parameters could modify the structure and viral capacity of CoVs, mainly SARS-CoV-2, in natural aquatic environments to infect aquatic biota.

Some CoVs transmitted through contaminated food or water include *Alphacoronaviruses* such as 229E and NL63 or *Betacoronaviruses* such as OC43 and HKU1 (not well-known frequencies) and SARS (with occasional frequencies) [6]. Infections related to contaminated water can have various routes of contagion that may include aspiration, inhalation of aerosol droplets, penetration by the skin or mucous membranes, as well as by intake [47]. For CoVs in general, it is reported that they can continue to be active or infectious for up to several weeks in water, including wastewaters [25]. Thus, viruses can come into contact with free-living aquatic organisms or be found in aquaculture farms and then be transmitted to humans when they become pets or food, and that, in the near future, it could be the way for new coronavirus outbreaks, which will need to be evaluated particularly for COVID-19.

## 2.2. CoVs and Aquatic Organisms

The aquatic biota is constituted by those organisms that inhabit aquatic environments temporarily or permanently. Within them, we find amphibians, mollusks, fish, and waterfowls. Studies focusing on viral infections in aquatic organisms have targeted mainly those species of commercial importance, especially those of aquaculture exploitation [46], or those associated with captivity and the tourism industry, such as marine mammals in aquariums and water parks [48]. However, aquatic biota includes various types of organisms. Although they do not have direct commercial or exploitative importance for man, they are part of aquatic ecosystems and are exposed to viral infections in this type of environment. Such is the case of amphibians such as frogs and salamanders or different aquatic birds such as gulls or pelicans.

Within the CoVs identified in aquatic organisms and associated with important pathological infections, we find those that affect fish, marine mammals, frogs (Table 1), and waterfowl (Tables 2 and 3; Figure 1). No mollusk and crustacean infections by CoVs have

been reported so far. However, a recent study detected SARS-CoV-2 non-infectious RNA (and highly degraded) in clams [49].

### 2.2.1. Fish

In the case of fish, viruses of the *Bafinivirus* group have been reported [7,14,50,51]. It is recognized that the first report of CoVs was in a European cyprinid known as white bream (*Blicca bjoerkna*), which showed a bacilliform structure related to the viruses of the subfamily Torovirinae, giving rise to a new genus defined as *Bafinivirus* [14]. Since the late 1980s, there have been cases where CoVs have been identified, particularly in Japan in 1988, in the common carp (*Cyprinus carpio*), widely cultivated and consumed in the world [52,53]. Production involvements reached 70% mortality within 20 days, with pathological damage including renal and hepatic tubular necrosis and damage to renal hematopoietic tissue [50], skin, and abdomen bleeding [14]. However, *Bafiniviruses*, together with *Gammacoronaviruses* detected in marine mammals, are usually found mainly in the digestive tract of the host or infected organisms [6]. Most recently, this genus has been reported in *Pimephales promelas* or bighead face fish [54,55] and in salmon, *Oncorhynchus tshawytscha* [56], identified as two different types of *Bafinivirus* (Table 1).

**Table 1.** Coronavirus found in aquatic organisms. Taxonomical groups, according to de Groot et al. [7] and Kasmi et al. [11].

| | Group | Genus | CoV Type | Host | Health Effects | Year | Reference |
|---|---|---|---|---|---|---|---|
| **Order Suborder Family** | Nidoviales Cornidovirinea Coronaviridae | *Coronavirus (?)* * | Carp CoV | Common carp (Japan) | Erythema, necrosis (abdomen and liver) Dermal ulcerations, necrotic lesion. Found in spleen and hematopoietic | 1988 | [51,57] |
| | | *Coronavirus (?)* * | Carp Viremia-Associated Ana-Aki-Byo | Common carp (Japan) | | 1997–1998 | [57,58] |
| **Subfamily** | Orthocoronavirinae | *Alphacoronavirus* | HSCoV | Harbor seals (Aquatic Park, FL, USA) | Acute enteritis, pulmonary edema | 1987 | [59] |
| | | | HSCoV | Pacific Harbor seals (CA, USA) | Pulmonary congestion, consolidation, and hemorrhage, pneumonia | 2000 | [60] |
| | | *Betacoronavirus* | Not reported | | | | |
| | | *Gammacoronavirus* | BWCoVSW1 | Beluga whale (Aquatic Park, CA, USA) | Hepatic necrosis and pulmonary disease | 2008 | [7,61] |
| | | | BdCoV HKU22 | Bottlenose Dolphin (Aquatic Park, Hong Kong) | Found in feces | ? | [10] |
| | | *Deltacoronavirus* | Not reported | | | | |
| | Torivirinae | *Torovirus (?)* * | CIVH 33/86 | Grass carp (Hungary) | Not known | 1986 | [51] |
| | | *Bafinivirus* | WBV DF24/00 | White bream fish (Saxony-Anhalt, Germany) | Not known | 2000 | [13,51] |
| | | | FHMNV | Fathead minnow fish (AR, USA) | Eyes and skin hemorrhage, tissue lesions (spleen, liver, and kidney) | 1997 | [54,55] |
| | | | Chinook Salmon Bfinivirus | Chinook salmon (ON, Canada) | Not known | 2014 | [51,56] |
| | Letovirinae | *Alphaletovirus* | Microhyla letovirus MLev | Ornamental pygmy frog | Not reported | 2018 | [12] |

* Still unclassified.

Some Chinese electronic media have stated that SARS-CoV-2 cannot be transmitted through fish, under the argument of the virus' thermo-tolerance and the low body temperature of fish compared to mammals [62]. However, this must be confirmed and might not be necessarily true. Recent and specific tests of SARS-CoV-2 tolerance to thermal gradients have demonstrated tolerance between 4 °C and 20 °C in the air [63]. This temperature range is below the human body temperature (36.6 °C), which is resisted and tolerated by SARS-CoV-2 during human infections. In drinking water, SARS-CoV-2 dispersion seems to decrease at 23 °C [64]. In addition, we must recognize that this would have relevant implications for viral tolerance if we consider that fish are ectothermic, that is, their body temperature is equal to the environment around them (except tuna, which is heterothermic) [65]. Fish that have already shown the presence of CoVs are tolerant to a wide range of temperatures, from temperate to warm conditions (like cyprinid fish that live or reproduce between 17 °C and 30 °C [66]. This aspect is essential in the case of SARs-CoV-2 because it might imply that under that temperature range, its incorporation by fish could be possible, only considering the body temperature as a limit factor for its infection. However, other defensive mechanisms could be involved. It is known that fish have an exclusive defense mechanism that consists of the secretion of mucus that completely covers the fish's body; this innate immune response, together with the ability to regulate different temperature ranges, prevents the replication and assembly of viruses [51].

On the other hand, marine mammals regulate body temperatures in a lower range than humans (29–32 °C [67]). Despite this, marine mammals have developed CoVs infections, as can be seen in Table 1. Under this evidence, the body temperature would not be a limiting factor on the possible future infection of SARS-CoV-2 to fish and marine mammals, or at least once they have already entered the body of the organism.

### 2.2.2. Waterfowl

Waterfowl is the group of animals most representative that can harbor CoVs [38] associated with aquatic environments. Birds appear to be the group with the highest diversity of CoVs, with at least 88 genetically identified varieties found during a recent review [68] and here updated (Tables 2 and 3), within two of the four genera (75 *Gammacoronavirus* and 13 *Deltacoronavirus*, respectively) of the subfamily Orthocoronavirinae [69]. Different types of birds, such as gulls (*Laurus hyperboreus*; *L. galucesceus*), geese (*Branta bernicla*, *Anser caerulescens*), spatulas (*Platalea minor*), herons (*Ardea cinerea*, *Ardeola bacchus*), cormorants (*Phalacrocorax carbo*), and ducks (*Anus Americana*, *A. crecca*, *A. clypeata*, *A. penelope*, *A. acuta*, *Dendrocygha javanica*) are confirmed carriers of CoVs. The ducks of the genus *Anas sp.* and *Anser* sp. are the most represented and capable of carrying even strains of SARS-CoV. Particularly noteworthy are *Anas domestica*, *Anas platyhynchos*, and *Anser anser* (Table 2).

**Table 2.** Coronavirus found in waterfowl of the order Anseriformes.

| Order | Genus | Species | Type of Bird | Type of CoV | References |
|-------|-------|---------|--------------|-------------|------------|
| Anseriformes | *Anas* | *domestica* | Duck | Gamma (SARS-CoV) | [70] |
| | | *platyhynchos* | Spot-billed duck | Gamma (SARS-CoV) | [70] |
| | | *americana* | American wigeon | Delta (JQ065048.1) | [71] |
| | | *crecca* | Common teal | Gamma (J0109, J0212, J0126, J0559, J0579, J1393); Delta (J1420) | [69,72,73] |
| | | *clypeata* | Northern shoveler | Gamma (K547, K554, K561, K589, J0554, J0807, J1300,J0901, J1491); Delta (J0590) | [69,73] |
| | | *penelope* | Eurasian wigeon | Gamma (K596, J0588, J1561) | [69] |

**Table 2.** *Cont.*

| Order | Genus | Species | Type of Bird | Type of CoV | References |
|---|---|---|---|---|---|
| | | *acuta* | Northern pintail | Gamma (J1375, J1393, J1404,, J1407, J1435, J1616, J1451,, PBA-10, PBA-15, PBA-16,, PBA-25, PBA-37, PBA-124) | [69,72,73] |
| | | *erythrorhyncha* | Red-billed duck | Gamma, (KM093874, KM093875,, KM093876, KM093877) | [73] |
| | | *hottentota* | Hottentot teal | Gamma (KM093880) | [71] |
| | *Clangula* | *hyemalis* | Long-tailed duck | Gamma (Fin14395) | [72] |
| | *Cygnus* | *cygnus* | Whooper swan | Gamma (Fin4983) | [72] |
| | *Dendrocygna* | *javanica* | Lesser whistling | Gamma (KH08-0852) | [69,71] |
| | | *viduata* | White-faced whistling duck | Gamma (KM093872, KM093873, KM093878) | [73] |
| | *Aythya* | *fuligula* | Tufted duck | Gamma (J1482) | [69] |
| | *Anser* | *caerulescens* | Snow goose | Gamma (WIR-159) | [69,71] |
| | | *anser* | Greylag goose | Gamma (SARS) | [70] |
| | | *cygnoides* | Swan goose | Gamma (DPV_16), Delta (DPV_5, DPV_10) | [71] |
| | *Branta* | *bernicola* | Brent goose | Gamma (KR-69, KR-70, KR88) | [69,71,73] |

**Table 3.** Gammacoronavirus and Deltacoronavirus reported in aquatic birds.

| Order | Genus | Species | Type of Bird | Type of CoV | References |
|---|---|---|---|---|---|
| Pelecaniformes | *Ardeola* | *bacchus* | Pond heron | Delta (KH08-1475, KH08-1474) | [69] |
| | *Ardea* | *cinerea* | Gray heron | Delta (K581, K513) | [69] |
| | *Bubulcus* | *ibis* | Heron | Gamma (KM093897) | [71,73] |
| | *Platalea* | *minor* | Black-faced spoonbill | Delta (J0569) | [69] |
| | *Phalacrocorax* | *carbo* | Great cormorant | Delta (J0982, J1517) | [69] |
| Gruiformes | *Rallus* | *madagascariensis* | Madagascar rail | Gamma (KM093896) | [73] |
| | *Porphyrula* | *alleni* | Allen's gallinule | Gamma (KM093890, KM093891, KM093892 KM093893, KM093894) | [71,73] |
| | *Gallinula* | *chloropus* | Common moorhen | Gamma (KM093881, KM093885, KM093887). Delta (JQ065049.1) | [71,73] |
| Charadriiformes | *Charadrius* | *pecuarius* | Kittlitz´s plover | Gamma (KM093879, KM093883, KM093884) | [71,73] |
| | *Gallinago* | *macrodactyla* | Madagascan snipe | Gamma (KM093888, KM093889, KM093895) | [73] |
| | *Calidris* | *mauri* | Western sandpiper | Gamma (KR-28) | [71] |
| | | *ptilocnemis* | Rock sandpiper | Gamma (CIR-66187, CIR-665821, CIR-665828) | [69,71,73] |

**Table 3.** *Cont.*

| Order | Genus | Species | Type of Bird | Type of CoV | References |
|---|---|---|---|---|---|
| | | *alba* | Sanderling | Gamma (PNLP100) | [71] |
| | | *fuscicollis* | White-rumped sandpiper | Gamma (PNLP159) | [71] |
| | | *sp* | Gull | Delta (JX548304) | [71] |
| | *Larus* | *argentatus* | Herring gull | Gamma (Fin9211, Fin10877, Fin10879, Fin12822, Fin13125) | [72] |
| | | *hyperboreus* | Glaucous gull | Gamma (PBA-173) | [69] |
| | | *fuscus* | Lesser Black-back gull | Gamma (Fin10059) | [72] |
| | | *glauscescens* | Glaucous-winged gull | Gamma (CIR-66002, GU396682) | [69] |
| | *Chroicocephalus* | *ridibundus* | Black-headed gull | Gamma (CIR-66187, GU396679, GU396680, GU396683, KX588674, Fin10083) | [69,72] |
| | *Rostratula* | *benghalensis* | Greater Painted-snipe | Gamma (KM093883) | [71] |
| | *Rynchops* | *niger* | Black skimmer | Delta (PNLP115) | [71] |
| Sphenisciformes | *Spheniscus* | *magellanicus* | Magellanic penguin | Avian CoV M41, C46, A99, JMK | [74] |
| | | *humboldti* | Humboldt penguin | Avian CoV M41, C46, A99 | [75] |
| | *Eudyptes* | *chrysocome* | Southern rockhopper | Avian CoV C46, A99, JMK | [76] |

A review carried out by Wartecki and Rzymski [38] highlighted that the prevalence of coronavirus in waterfowl varies according to the geographical region, from a low incidence in South America (Brazil) up to 19% in Scandinavia and that their infections can range from asymptomatic to mild or severe. Waterfowl feces deposited directly in the water are considered a critical CoVs source to aquatic environments, contributing to viral spread, an idea that is still to be confirmed [38]. The contribution of waste and carcasses from bird clutches to water bodies where they live allows CoVs to enter the aquatic food chain and can then be distributed with the potential of infection to other organisms or even compromise food quality as avian influenza [38,77].

Recently, in evaluating infectious agents in the penguin *Spheniscus magellanicus*, four variants of CoVs (Avian coronavirus M41, C46, A99, and JMK) were detected, with ACoVC46 being the one with the highest representation, causing bronchitis [74]. The seroprevalence analysis in penguins, which evaluates the percentage of the population that has developed antibodies against CoVs, shows a low prevalence in two more known species as *Eudyptes chrysocome* (Southern Rockhopper) [76] and *Spheniscus humboldti* (Humboldt penguin) [75], respectively.

Some birds, such as cormorants and ducks, are migratory, and that would allow wide geographical distribution of these types of viruses. Although no reports of human infection originated from waterfowl have been detected [38], the ecological study of these correlations becomes indispensable to better understand the relationship of birds and CoVs, and their epidemiology among birds and other species within their ecosystems [69]. It has been suggested that Deltacoronaviruses, through recombination processes of their genome, could lead to their propagation through new hosts [38].

### 2.2.3. Amphibians

Recent research on the nido-like virus by Bukhari et al. [12] has revealed sequence encoding proteins similar to proteinases of the coronavirus. Through these analyses, the authors discovered an RNA transcript of 22.3 Kb from nine tadpoles belonging to *Microphyla fissipes* (ornamented pygmy frog), the only amphibian host reported so far (Table 1), to the best of our knowledge. This virus has been named *Microphyla letovirus* (MLeV), which was included in the nidovirus taxonomy list published by the International Committee on the Taxonomy of Viruses in 2018 [12].

### 2.2.4. Marine Mammals

Regarding marine mammals, the first report of CoVs dates back to the 1970s associated with the death of several seals (*Phoca vitulina*) in a Florida aquarium [59] and other free-living pinnipeds off the coast of California [78]. This infection was known as HSCoV (harbor seal coronavirus), identified as deadly hemorrhagic pneumonia caused by the *Alphacoronavirus* group [78]. Years later, in 2008, the presence of other CoVs was detected in a beluga whale *(Delphinapterus leucas)* under captivity (BWCoV SW1) [10,61]. In 2014, the presence of CoVs was detected in feces from bottlenose dolphins [10], from the Indo-Pacific (*Tursiops aduncus*), which was called BdCoV HKU22 (bottlenose dolphin CoV). These latter two were recognized within the *Gammacoronavirus* group, which caused viral bronchitis to those infected animals [10]. The gregarious behavior of several marine mammal species may promote the contagion and dispersal of these types of pathogens in wild populations, so their monitoring becomes necessary for their health and avoids further transmission to other aquatic organisms.

Gammacoronavirus detected in marine mammals, unlike *Bafinivirus* in fish, can also be found in the respiratory tracts of terrestrial and marine mammals and not only in the digestive tracks [6]. Viruses such as influenza A and B have been reported in mammals such as seals and cetaceans [6,48], considered reservoirs and vectors towards humans [6]. This scenario opens the possibility that other viruses, including CoVs, could be transmitted to humans when interacting and coming into contact with seals, sea lions, and dolphins in water parks and aquariums. Working with infected wildlife, or using them as food sources, especially in communities such as Asian ones, might also be another route of transmission.

### 2.2.5. Crustaceans and Mollusks

Crustaceans are recognized as a group capable of accumulating human pathogens in their exoskeleton or body cover [79]. However, no coronavirus infections have been reported to the best of our knowledge. Notwithstanding, in shrimp, the "yellowhead virus", or YHV, [61] was reported in 1990 in East and Southeast Asia, affecting farmed *Penaeus monodon* shrimp. There are unconfirmed reports in Mexico for *P. Stylirostris* and *P. vannamei* [80–82]. This virus belongs to another family of Nidovirales, known as Roniviridae, specifically the *Okavirus* genus [11]. Although it does not belong to the Coronaviridae family, it is a closely related virus, which illustrates the probability that it may also be possible to find CoV infections in crustaceans in the future. Genetic studies have shown that YHV had undergone significant recombination processes, apparently attributable to international trade with wild and farmed shrimp in the Asia-Pacific region, promoting a faster genetic diversity of the virus due to several recombination events [83]. So, the possibility cannot be ruled out that type of event is taking place in the wild for coronavirus species.

In mollusks, until recently, there were no reports of the presence of CoVs in scientific literature. However, Polo et al. [49] detected virions or RNA fragments from SARs-CoV-2 in clams (*Ruditapes phiippinarum* and *R. decussatus*), collected between May and July 2020 in Galicia, Spain, the first report of detection of this coronavirus in coastal environments and a reflection of contact with aquatic fauna. The authors emphasize that although this presence reflects that SARS-CoV-2 can reach the benthic marine fauna, under certain circumstances, the presence of RNA fragments of the virus itself does not represent a risk of infection for both mollusks and humans, especially through digestion [49]. However,

the authors highlight the importance of taking precautionary and monitoring measures in aquatic environments, including using bivalve mollusks as bioindicators, allowing us to understand better their presence, persistence, and risks for aquatic biota [49].

There are reports of RNA virus infections, both encapsulated and unencapsulated. Arenaviridae or Retroviridae (encapsulated) families, as well as Biriaviridae, Reoviridae, and Picornaviridae (unencapsulated), have been found in bivalves [57].

As mentioned in Section 1, CoVs are RNA genome viruses with a protein cap, a feature shared with the Arenaviridae and Retroviridae families (all three are RNA-encapsulated viruses). As in the case of crustaceans, it is possible that in the future, similar RNA encapsulated viruses such as CoVs could develop infection patterns in mollusks, similar to those observed, for example, in the cases of *Hyriopsis cumingii* plague virus (HcPV, Arenaviridae) or B-type retrovirus (Retroviridae), the first one reported in the freshwater mussel *H. cumingii*, and the second in the marine clam *Mya arenaria* [57].

It is also impossible to rule out that future environmental and genetic changes may cause new routes of spread and transmission of coronavirus species through vector organisms such as crustaceans and mollusks. The Arenaviridae viruses are considered enteric viruses of interest for the WHO (World Health Organization); they are rarely detected (low frequency) in water or food during human transmission studies [6]. Nevertheless, some arenavirus infections, such as Lassa fever, caused by the *Lassa mammarenavirus* and transmitted by mice to humans, are considered to have the potential to increase their transmission under climate change scenarios [84]. The same case might apply to all viruses, including Retrovirals, which are usually associated with swimming pools [14], and fish infections [13,14].

Analyzing the different aspects related to the presence of coronavirus in the above-mentioned aquatic organisms, especially the detection of its presence in marine mammals such as dolphins and seals, leads us to one of the main questions in case of the SARS-CoV-2 infection of an aquatic ecosystem. Would SARS-CoV-2 from aquatic organisms infect humans? Their biological similarities with other viruses detected in other groups of animals such as crustaceans or mollusks (whose presence has been associated with transfers to humans due to their use as food).

To the best of our knowledge, the spike-binding proteins of the SARS-CoV-2 coronavirus show a high specificity for the ACE2 membrane receptors (angiotensin-converting enzyme 2, which is usually found in different types of cells and tissues, including the epithelial cells of the nose, mouth, and the lungs in humans). However, these same receptors have also been found in finfish with 59% amino acid sequence homology compared to human receptor protein [85–87]. Some authors point out that coronaviruses prevail in cyprinids naturally due to the fact that these viruses have been isolated from free dead carp, goldfish, and crucian carp (*Carassius carassius*). Their clinical effects include reducing food intake, pseudo-feces, petechiae, and increased mucus accompanied by mortality, according to Fichtner et al. [88] (cited in Al-Taee et al. [85]). Sano et al. [50] stated that carp coronaviruses could replicate at a temperature of 20 °C and be transmitted in water. Thus, the contact of unprotected personnel working on aquaculture farms with animals or carp tissues of these species could lead to potential infections of other types of coronaviruses in humans in the future.

Some aquatic organisms can potentially become infected with SARS-CoV-2 under certain conditions. According to a genomic study by Damas et al. [89], who compared in 410 species of vertebrates the presence of the ACE2, a long list of species vulnerable to being infected by SARS-CoV-2 were found, among these being some marine mammals such as grey whales and bottlenose dolphins with a high vulnerability. In contrast, for some birds (n = 72), fish (n = 65), amphibians (n = 4), and reptiles (n = 17), they found a very low vulnerability to contracting SARS-CoV-2. The previous study helps to know which animal populations (particularly those with the ACE2 receptor enzyme in their tissues) can be affected by the SARS-CoV-2 virus and the risks for these species. It is highly recommended

to extend these studies to identify whether the ACE2 protein is found in freshwater and marine aquatic invertebrates.

### 3. Some Viral Infections to Humans from Aquatic Organisms

No published studies on the actual risk of SARS-CoV-2 contagion from aquatic organisms were found during this review. Neither have *Betacoronavirus* (genus to which SARS-CoV-2 belongs) been found in marine organisms; instead, it has been seen those other genera such as *Gammacoronavirus* (which were discussed in Section 2.2.4) and *Alphacoronavirus* prevail, which share little homology with the SARS-CoV-2 virus. However, according to Mordecai and Hewson, these two genera are associated with respiratory diseases in pinnipeds and cetaceans [90].

There is a history of other viral respiratory infections transmitted to humans from wild or captive animals [19,48]. That is the case with influenza-A, caused by the H7N7 virus, in people infected during a necropsy performed to a seal [59] or by coming into contact with the sneeze of a seal in captivity [48], causing conjunctivitis, rather than typical influenza or respiratory disease. A similar case has also been identified for Influenza B [6,48]. Moreover, a historical review carried out by Petrovic et al. [19] has shown numerous viral outbreaks (not CoVs related) associated with shellfish. These outbreaks included human enteric viruses, mainly those of type NoV (norovirus). HAV (hepatitis virus A), EV (Enterovirus), HAdV (human adenovirus), and HRV (human rotavirus) are reported in shellfish in different countries, but not CoVs. Oysters and clams have been associated with NoV and HAV between 1976 and 1999 in the United States alone. These viruses have also been identified in mollusks in Europe, both in fish and sea markets and oyster farms associated with human enteric viruses between 1990 and 2006 [91,92]. All are good examples of the food source of viral infections. For the World Health Organization and the Food and Agriculture Organization Joint Committee, coronaviruses related to Severe Acute Respiratory Syndrome (SARS-CoV) are viruses of concern by contaminated food [93]. Other types of water viruses associated with birds, such as H5N1 avian influenza and avian influenza A1, are highly infectious and recognized for their transmission to humans from duck meat and blood [94,95]. Due to these examples, extensive monitoring studies are required since ducks are one of the main groups of birds capable of carrying CoVs (Table 2).

At the moment, as long as there are no more significant scientific elements to be certain of the non-spread of the SARS-CoV-2 pandemic through natural waters and aquatic organisms, it is best to follow the indications that the health authorities have been issuing in this regard. These indications highlight those made by the World Health Organization [96], which recommends avoiding unprotected contact with wild and farm animals, and has even been recommended not to approach public markets where wild animals are under sale, both live and slaughtered [9].

### 4. Further Considerations about CoVs and Aquatic Biota

The efforts of the scientific community will continue over the coming years to learn more about COVID-19. Studying genetic adaptation, including mutation and recombination, identifying routes of zoonotic (animal) origin, new vector organisms (birds, mammals, fish, amphibians, mollusks or crustaceans), animal-human transmission events, wild natural storage, and contagion risks, will allow effective and realistic programs to control the transmission of coronaviruses, particularly SARS-CoV-2. It is recognized that viral genotypes with epidemiological potential can become exceedingly variable due to their genetic characteristics, which allow them to endure and survive and spread and even mutate along trophic chains [79].

As some studies suggest, the relationship between coronaviruses and aquatic biota is not limited to a pathogen–host relationship but also a pathogen–treatment relationship.

There is a universe of biologically active substances of marine origin, such as flavonoids, phlorotannins, alkaloids, terpenoids, peptides, lectins, polysaccharides, lipids, and other

substances that can affect coronaviruses (Table 4). The penetration and entry stages of the viral particle into the host cell in viral nucleic acid replication and virion release from the cell can also act on the host's cellular targets. These natural compounds could be a vital resource in the fight against coronaviruses [97]. Zaporozhets and Bedsenova [97] conducted a database search in 2020, identifying ~34 biologically active substances from sponges, echinoderms, mollusks, soft corals, bryozoans, and others.

It is encouraging to know that even other aquatic organisms, such as seaweed or sponges, could play a key role in treating CoVs infections. It has been observed through laboratory tests with *Halimeda tuna* algae, a natural product known as diterpene aldehyde or halitunal [98], an antiCoV effect. Other examples are the sponge *Mycale* sp., which produces a substance called micalamide A, both with antiviral capacity against the A59-CoV of murine origin [99,100].

**Table 4.** Bioactive substances of marine origin have an antiviral activity to CoV and other viruses.

| Substances (Compound) | Species | Source | Reference |
|---|---|---|---|
| Peptides (Pseudoteonamides C y D) | *Theonella swinhoei* | Marine sponge | [101] |
| Nucleosides (micalamide A) | *Mycale* sp, | Marine sponge | [100] |
| Nucleosides (Spongotimidine and Ara-A9 | *Cryptotethya crypta* | Marine sponge | [102] |
| Esculetin ethyl ester | *Axinella corrugata* | Marine sponge | [103] |
| Alkaloid (Dragmacidine F) | *Halicortex* sp. | Marine sponge | [102] |
| Alkaloid (4-methylaptamine) | *Aaptos aaptos* | Marine sponge | [102] |
| Terpenic aldehydes (halitunal) | *Halimeda* sp. | Tuna seaweed | [99] |
| Florotannins (Dieckol,, 6,6'-Bieckol, 8,8'-Bieckol) | *Ecklonia cava* | Brown seaweed | [104] |
| **Florotannins** (Heptafuhalol A, Phlorethopentafuhalol A, Pseudopentafuhalol B, Pseudopentafuhalol C, Hydroxypentafuhalol A | *Sargassum* sp. *Sargassum spinuligerum* | Brown seaweed | [104] |
| Fatty acids (oleic acid) | *Ceramium virgatum* *Ulva intestinalis* *Fucus* sp. | Red algae Green algae Brown seaweed | [105] |
| Phytosterols (Saryngosterols) | *Acanthophora spicifera* *Cladophora fascicularis* *Sargassum muticum* | Red algae Green algae Brown seaweed | [105] |
| Phytosterols (β-sitosterol) | *Euchema cottonii* *Ulva fasciata* *Sargassum glaucescens* | Red algae Green algae Brown seaweed | [105] |
| Glycoglycerolipids | *Exophyllum wentii* *Sargassum horneri, Phormidium* sp. | Red algae Brown seaweed, Cyanobacteria | [105] |
| Kjellmanianone | *Sargassum naozhouense* | Brown seaweed | [105] |
| Loliolide | *Sargassum naozhouense* | Brown seaweed | [105] |
| Caulerpin | *Caulerpa racemosa* *Chondria armata* *Sargassum platycarpum* | Green algae Red algae Brown seaweed | [105] |

**Table 4.** *Cont.*

| Substances (Compound) | Species | Source | Reference |
| --- | --- | --- | --- |
| Polyhydroxy naphthoquinones natural pigments (Echinochromo A, Echinaminas A and B, beta-carotene, Astaxanthin and Fucoxanthin) | *Scaphechinus mirabilis* *Strongylocentrotus nudus* *S. pallidus* *S. polyacanthus* *Echinarachnius parma* *Evechinus chloroticus* *Echinometra mathaei* *Arbacia dufresnii* | Sea urchin (gonads, spines and carapace) | [106–108] |

Another good example is the *Axinella corrugata* sponge that produces an ethyl ester of esculetin-4-carboxylic acid against SARS-CoV [103]. Natural marine compounds are being found to be inhibitors against the main protease of SARS-CoV-2 [109]. Eight antiviral compounds have been recently reported against SARs-CoV-2 protease, particularly a compound identified as Esculetin ethyl ester from *A. corrugata*, which has shown to be the most effective antiprotease [110]. These molecules are considered bioactive compounds that act as replication inhibitors to SARS-CoV in Vero cells (kidney cells from an African green monkey [103], among others that can act as effective antiviral drugs [110]. Other SARS-CoV-2 protease inhibitor compounds are pseudotheonamides, which have been isolated from the marine sponge *Theonella swinhoei* and have shown good inhibitory activity on serine protease [97]. Together with other products of natural origin [98,103], these substances could be the source of some control against coronavirus like SARS-CoV-2 in the future.

One species that has shown significant therapeutic potential against SARS-CoV-2 are sea urchins. The consumption of these organisms dates back to ancient times in Chinese medicine. Gonads, spines and shell powders are known for their beneficial effects on the heart, bones, blood, and impotence [111]. In the 1980s, Echinochroma A (polyhydroxy naphthoquinone molecule) from gonads, spines and shells showed cardioprotective action and healing properties for some eye diseases [111]. Recently, the antioxidant and antiviral activity of Echinochroma A against RNA viruses such as TVEB (tick-borne encephalitis virus) and DNA virus, HSV-1 (herpes simplex virus type 1) are being investigated with good results to develop drugs [107]. The antiviral effects of Echinochroma A that of its amino homologues, Echinamines A and B on the HSV-1 virus, have shown that they exert a response that inhibits the production of reactive oxygen species generated by HSV -1; in addition, these molecules reduce the adhesion of the virus to the host cell [106]. Recently Barbieri et al. [112] have evaluated the therapeutic potential of sea urchin pigments (Echinochroma A, Spinochromos A and B) on SARS-CoV-2, observing that it acts by blocking and inhibiting the protein S of the virus, which prevents it from entering the host cell. Rubilar et al. [113] evaluated the potential of the pigments (Spinochromo A, Echinochroma A, beta-carotene, Astaxanthin and Fucoxanthin). They found an inhibitory capacity and high affinity to the SARS-CoV-2 protease (which plays an active role in its replication process) from the Echinochroma A pigment. This result suggests that sea urchin pigments may be candidates for being antiviral drugs against the SARS-CoV-2. A disadvantage is that sea urchin pigments are found in low concentrations; however, there is high concentration of this pigment in eggs of the sea urchin *Arbacia dufresnii* [113].

Marine macroalgae, for example, those of the genus *Sargassum* sp. constitute a promising source of compounds with antiviral activity, motivating the search for new drugs against viruses [114]. Gentile et al. [104] carried out a promising and cutting-edge study where different compounds of natural origin with inhibitory activity for the SARS-CoV-2 protease have been selected, modelled, molecularly simulated, and evaluated from the data of a library of 3D chemical structures of natural marine products. According to the authors, the most promising SARS-CoV-2 inhibitors are mainly related to phlorotannins, oligomers of phloroglucinol (1,3,5-trihydroxy benzene), isolated from *Sargassum spinuligerum*. Phlorethols, fuhalols, and fucophlorethols are among phlorotannins found in

other species of *Sargassum* [104]. It was also observed that the most active inhibitor compounds against SARS-CoV-2 are those belonging to the family of phlorotannins, isolated in the brown alga *Ecklonia cava*, which is an edible alga recognized as a rich source of bioactive derivatives [98].

Ponce Rey et al. [114] evaluated the in vitro antiviral activity of a hydroalcoholic extract of the brown seaweed *Sargassum fluitans* against Echovirus 9 (E9). This type of enterovirus causes severe systemic diseases such as aseptic meningitis and, to a lesser extent, infant mortality. There is no antiviral treatment or vaccination against this virus. As in most enteroviruses, the phytochemical screening showed quinones, proanthocyanidins, catechins, polar triterpenes, hydrolysable tannins, and proteins as main constituents. The extract was not cytotoxic at the concentrations evaluated, and it potently inhibited the replication of E9 in the cell line used with a high SI (selective index) of 95.05. Also, during the antiviral test, the hydroalcoholic extract of *S. fluitans* inhibits E9 multiplication in a dose-dependent manner [114].

Abdelrheem et al. [105] evaluated the inhibitory effect of the SARS-CoV-2 virus protease by different natural bioactive compounds with recognized biological activity as anticancer, anti-inflammatory, antimicrobial, antiviral, among others. Some examples are hexadecanoic acid, oleic acid, saryngosterols, beta-sitosterol, glycoglycerolipids, Kjellmanianone, the terpene loliolide and the alkaloid caulerpin. These bioactive compounds are found in green, red, and brown macroalgae of marine origin (*Sargassum platycarpum S. naozhouense*, *S. horneri*, *S. glaucescens*, *S. muticum*, *Ceramium virgatum*, *Fucus* sp., *Cladophora fascicularis*, *Acanthophora spicifera*, *Ulva fasciata*, *U. intestinalis*, *Eucheuma cottonii*, *Exophyllum wentii*, *Chondria armata*, *Caulerpa racemosa*), including freshwater microalgae (*Chlorella vulgaris*, *Nannochloropsis*). The most outstanding results of the study allow us to conclude that caulerpin is very effective against SARS-CoV-2 by inhibiting the main protease and can be further explored for drugs against the COVID-19 pandemic [105].

Some issues faced by those developing new drugs to inhibit the SARS-CoV-2 and other viruses from natural products is that sources are not readily available or cannot be developed on a large scale. In this regard, it is worth mentioning that some coastal areas of the Caribbean Sea, as well as the coasts of Mexican state of Quintana Roo, from 2011 up to date, are facing massive *Sargassum fluitans* and *S. natans* arrivals. Other countries that have reported massive waves of *Sargassum* are Belize, Honduras, Jamaica, Cuba and Barbados [115]. This type of macroalgae proliferates, doubling its mass in less than 18 days. These algae eventually decompose and generate unpleasant odors and cause problems in coastal areas and beaches. It seems that *Sargassum* can interfere massively in the transmission of light down the water column, mainly affecting seagrasses. When *Sargassum* dies and decomposes, it consumes large amounts of oxygen, causing anoxia, affecting other species [115]. Therefore, it would be interesting to evaluate whether the *Sargassum* that reaches the shores can serve as a potential source of bioactive compounds (antivirals, antioxidants, among others) that can be used to combat SARS-CoV-2 or other viruses.

## 5. Conclusions

CoVs in aquatic environments are rare but also a reality, which has demonstrated its ability to be transmitted to organisms in wildlife, aquaculture farms, and animals under captivity. Its presence observed in farmed fish such as carp, or a frog, although they have not reported significant effects or consequences on human health, could be of potential risk in the near future for aquatic ecosystems. Knowledge of other cases such as marine mammals have shown to be carriers of respiratory infections, must be considered for other viruses like CoVs, particularly SARS-CoV-2. Waterfowl show to be a natural CoVs reservoir, mainly ducks, which deserve to be studied in more detail due to their migratory behaviors. This monitoring should be carried out jointly with new studies and biotechnological strategies to continue searching for alternative bioactive compounds of aquatic origin that can be used against the COVID-19 pandemic, among other diseases.

**Author Contributions:** Conceptualization, G.N.-N. and design G.N.-N., J.A.V.-Á., A.T.-G.; Methodology, G.N.-N., J.A.V.-Á.; Figure, J.A.V.-Á.; Formal Analysis and Investigation, J.A.V.-Á., A.T.-G., A.A.G.-B., E.R.-A., F.A.Z.-G. and G.N.-N.; Writing—Original Draft Preparation, G.N.-N., J.A.V.-Á.; Writing—Review & Editing, G.N.-N., J.A.V.-Á., A.A.G.-B.; Funding, A.T.-G. All authors have read and agreed to the published version of the manuscript.

**Funding:** This work received no external funding, and Manzanillo University of Technology, Colima, Mexico funded the APC.

**Institutional Review Board Statement:** Not applicable.

**Informed Consent Statement:** Not applicable.

**Data Availability Statement:** Data sharing not applicable.

**Acknowledgments:** The author G.N.-N. recognizes the support provided by the PII SNI-UJAT program, comment, ideas and suggestions made by all reviewers and the assistance of Enrique Núñez-Jiménez and Elia Cornelio to improve this manuscript.

**Conflicts of Interest:** The authors declare no conflict of interest.

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
