# Peer review of "Aquatic Biota Is Not Exempt from Coronavirus Infections: An Overview"

_water, doi:10.3390/w13162215_

Round 1

Reviewer 1 Report

The authors describe the presence of various coronaviruses, including SARS-Cov2, among aquatic organisms, in light of some cases of viral infections transferred to humans from the aquatic biota. Starting from the environmental conditions in the water that allow the survival of different coronaviruses, the authors describe which aquatic animal species are receptive to viruses and whether they are capable of infecting humans. The study also suggests some other aquatic animal species such as algae and aquatic invertebrates as a source of cure or treatment.

Overall the manuscript is well organized, quite extensive.

I would just suggest replacing or adding a couple of recent specific articles about Covid-19 infection as follows:

1.Introduction, line 39: literature needs to be added: eg: Costagliola A, et al. Do animals play a role in the transmission of severe acute respiratory syndrome Coronavirus-2 (SARS-CoV-2)? A comment. Animals (Basel). 2020 Dec 24; 11 (1): 16. DOI: 10.3390 / ani11010016. PMID: 33374168; PMCID: PMC7823338.

  1. Coronavirus and aquatic environments, line 95 (ref 14), line 113 (ref 19), and line 175 (ref 39) could be replaced by the following paper: Aguiar-Oliveira ML, et al. Wastewater Based Epidemiology (WBE) and Viral Detection in Polluted Surface Waters: A Valuable Tool for COVID-19 Surveillance: A Brief Review. Int J Environ Res Public health. 2020 Dec 10; 17 (24): 9251. DOI: 10.3390 / ijerph17249251. PMID: 33321987; PMCID: PMC7764684, which extensively describes the presence of SARS-Cov2 in polluted waters from different countries, their supposed danger to public health, and environmental surveillance.

Author Response

Reviewer 1 comment: I would just suggest replacing or adding a couple of recent specific articles about Covid-19 infection as follows: 1.Introduction, line 39: literature needs to be added: eg: Costagliola A, et al. Do animals play a role in the transmission of severe acute respiratory syndrome Coronavirus-2 (SARS-CoV-2)? A comment. Animals (Basel). 2020 Dec 24; 11 (1): 16. DOI: 10.3390 / ani11010016. PMID: 33374168; PMCID: PMC7823338. 2.- Coronavirus and aquatic environmentsline 95 (ref 14), line 113 (ref 19), and line 175 (ref 39) could be replaced by the following paper: Aguiar-Oliveira ML, et al. Wastewater Based Epidemiology (WBE) and Viral Detection in Polluted Surface Waters: A Valuable Tool for COVID-19 Surveillance: A Brief Review. Int J Environ Res Public health. 2020 Dec 10; 17 (24): 9251. DOI: 10.3390 / ijerph17249251. PMID: 33321987; PMCID: PMC7764684, which extensively describes the presence of SARS-Cov2 in polluted waters from different countries, their supposed danger to public health, and environmental surveillance.

Authors' response: The authors appreciate the comments and suggestions provided by the reviewer. Both suggested references are undoubtedly enriching and provide support for the ideas raised in the text. Both references have been incorporated (see lines 38, 110, 128 and 190).

Reviewer 2 Report

Manuscript No.:            water-1322216

Title:                            Aquatic biota is not exempt from coronavirus infections: an overview

Authors:                       Smith et al

This is a concise and timely review.  The premise is quite interesting, especially these days when we all have been inundated with so many original articles and reviews related to SARS-CoV-2/COVID-19.  Having said that, the manuscript at times reads more like a collection of ideas with limited flow, cohesiveness.  The English grammar could also be improved.

Minor Comments:

  • There are many sections, paragraphs where sentences and ideas lack support, i.e., missing references.
  • Please review the use of COVID-19 vs SARS-CoV-2. As the authors stated, one is the disease and the other is the actual virus.  For example, in line 244: “ …COVID-19 cannot be transmitted…” - I’m sure the author meant SARS-CoV-2.
  • Finally, although interesting, the last section describing the potential use of natural products of marine, aquatic origin as antivirals is out of place. I would suggest to remove it from the review.

Author Response

Reviewer 2 comments: This is a concise and timely review.  The premise is quite interesting, especially these days when we all have been inundated with so many original articles and reviews related to SARS-CoV-2/COVID-19.  Having said that, the manuscript at times reads more like a collection of ideas with limited flow, cohesiveness.  The English grammar could also be improved.

Authors' response: The authors appreciate the opinions, comments and suggestions provided by the reviewer. As you can see in the revised version, English has been improved, trusting that it clarify the ideas and facilitates fluency in their reading.

Reviewer 2 comments: There are many sections, paragraphs where sentences and ideas lack support, i.e., missing references.

Authors response: Thank you for your comment. It will be advantageous to know on time those paragraphs and sections in particular. In the same way, we have made a general inspection trying to identify them, to give them the necessary support.

Reviewer 2 comments: Please review the use of COVID-19 vs SARS-CoV-2. As the authors stated, one is the disease, and the other is the actual virus.  For example, in line 244: "…COVID-19 cannot be transmitted…" - I'm sure the author meant SARS-CoV-2.

Author's response: Absolutely right. It has been corrected accordingly (see line 283).

Reviewer 2 comments: Finally, although interesting, the last section describing the potential use of natural products of marine, aquatic origin as antivirals is out of place. I would suggest to remove it from the review.

Author's Response: Thanks for your suggestion. Although it is true, the manuscript focuses on the fact that coronavirus infections have reached aquatic organisms; the authors consider keeping this section. Aquatic organisms can also be sources of possible controls or treatments against coronavirus. Either by the very nature of the biomolecules that they produce or perhaps as a result of mild infections. This information complements the relationship between coronavirus and aquatic biota; we must see it as a pathogen-infectious and as a pathogen treatment. A line has been added to clary it (see line 502).

Reviewer 3 Report

Over all the paper is highly informative, it would be nice if author can add some meaningful figure in the literature review. Also, please add methodology in detail.

Author Response

Reviewer 3 comments: Over all the paper is highly informative, it would be nice if author can add some meaningful figure in the literature review. Also, please add methodology in detail.

Author's response: We welcome comments and suggestions. We have added figure 1 (See page 6), which illustrates the main groups of coronaviruses and direct or indirect transfer routes towards aquatic environments and their fauna. The general procedure was added for document selection (See lines 63-75).